# Ecological Pest Control in Alpine Ecosystems: Monitoring Asteraceae Phytophages and Developing Integrated Management Protocols in the Three River Source Region

**DOI:** 10.3390/insects16080861

**Published:** 2025-08-19

**Authors:** Li-Jun Zhang, Yu-Shou Ma, Ying Liu, Jun-Ling Wang

**Affiliations:** Qinghai Provincial Key Laboratory of Adaptive Management on Alpine Grassland, Key Laboratory of Superior Forage Germplasm in the Qinghai-Tibetan Plateau, Academy of Animal and Veterinary Sciences, Qinghai University, Xining 810016, China; 13109728505@163.com (L.-J.Z.); 18293146085@163.com (J.-L.W.)

**Keywords:** the Three River Source Region (TRSR), *Aster* spp., arthropods, *Tephritis angustipennis*, pest control, integrated pest management (IPM)

## Abstract

*Aster* species are vital for restoring alpine grasslands on the Qinghai–Tibet Plateau, but they suffer significant pest damage, especially during flowering. This study identified key pests (like *Tephritis angustipennis*) and developed an environmentally friendly integrated pest management (IPM) strategy. The IPM approach includes physical barriers (flower bagging) and attractants to reduce pests. Targeted chemical spraying is used at specific plant growth stages for effective pest control with reduced resistance risk. Utilizing natural enemies (like spiders) helps control pest populations. This four-pronged IPM strategy enhances control efficiency while being eco-friendly, offering a practical solution for alpine grassland restoration and sustainable medicinal plant management in high-altitude areas.

## 1. Introduction

The Qinghai–Tibet Plateau (TP), the world’s highest and youngest major plateau [1,2,3], holds a significant position in global ecosystems due to its ecological fragility and biodiversity [4,5]. The Three River Source Region (TRSR), a World Heritage Site on the TP, features a unique alpine ecosystem primarily constructed around *Aster* spp. This genus is not only characterized by its high species diversity, widespread distribution, and varied uses but is also extensively employed in ecological restoration and grassland management due to its strong ecological adaptability and environmental tolerance [6]. However, insect herbivory (particularly during flowering and seed development stages) has emerged as a major threat to the conservation of its genetic resources, severely impacting seed yield and germplasm collection efforts [7]. Consequently, comprehensive surveys and targeted research on pests, especially key pests affecting *Aster* spp. in the TRSR, are urgently needed.

Current research focuses on the response mechanisms of *Aster* species to abiotic stresses, such as the antioxidant enzyme system under drought stress [8,9]. However, field surveys indicate that these plants are currently under attack by pests such as grasshoppers, aphids, and fruit flies [10]. These pests reduce yields by feeding on floral receptacles and seeds, severely limiting the application of *Aster* species in ecological restoration and medicinal cultivation [7,10]. While extensive taxonomic and ecological studies on *Aster* species exist [11,12], significant gaps remain in understanding plant–insect interaction mechanisms and pest management strategies in alpine environments. Against this backdrop, further investigations into the insect diversity of *Aster* spp. in the TRSR are needed, with a focus on population dynamics of key pests. Furthermore, developing effective, environmentally friendly, and sustainable integrated pest management (IPM) strategies is crucial for mitigating the harm caused by these pests to *Aster* species.

In recent years, the damage caused by fruit fly pests to agriculture, forestry, and grasslands has garnered increasing attention [13], with monitoring and control techniques demonstrating a diversifying trend. In terms of control methods, traditional chemical control, due to its low efficiency and environmental pollution, is gradually being replaced by novel monitoring technologies (e.g., olfactometers + smart sensors) and physical methods (e.g., flower head bagging, sex pheromones, and sterile insect technique) [14,15,16]. Biological control methods, including parasitoid wasps (*Fopius arisanus*), predatory natural enemies, plant essential oils, and microbial agents (such as *Bacillus thuringiensis* (BT)), have also made significant advancements [17,18]. Although chemical control still holds considerable importance [19,20], research on specific ecosystems such as the TP grasslands remains largely inadequate. Furthermore, existing studies indicate that the unique environment of the TP presents dual challenges for chemical control: degradation of efficacy due to ultraviolet radiation and ecological risks [21]. This highlights that the effectiveness of physical control methods under conditions of low oxygen and extreme temperature fluctuations also requires validation [22]. Additionally, the co-evolutionary mechanisms between the chemical defence systems of *Aster* species and their insect detoxification systems remain unclear, severely limiting the innovation and development of sustainable control strategies [7]. Key scientific questions include the following: How do altitude gradients influence the co-evolution of phenolic metabolism in *Aster* species and insect detoxification enzyme systems? Can integrated control measures effectively suppress pests while maintaining plant physiological homeostasis? These represent significant, overarching questions for future research. Our study contributes by specifically investigating population diversity, population dynamics, physiological responses, and the IPM interventions within our defined scope.

This study aims to establish a three-dimensional analytical framework integrating ecology, physiology, and control technologies to systematically investigate the *Aster*–insect interactions at different altitude gradients in the TRSR. Specific objectives include (1) characterizing the insect diversity and population dynamics affecting *Aster* spp. in the TRSR; (2) evaluating the comprehensive effectiveness of integrating chemical control with physical measures against the main pest (*Tephritis angustipennis*); and (3) elucidating the interaction mechanisms between defence-related enzymes in *Aster* species and key detoxification enzymes in insects. This study employs ecological surveys, physiological measurements, and control assessments to establish the IPM system, thereby providing theoretical and practical support for the sustainable utilization of biological resources in alpine ecosystems [23]. This work will not only directly fill existing research gaps but also offer critical theoretical foundations and practical guidance for the sustainable utilization of biological resources in high-altitude ecosystems.

## 2. Materials and Methods

### 2.1. Study Area and Survey Design

This study was conducted between June 2023 and September 2024 at the TRSR of Qinghai Province, China (altitude: 2303–4115 m, Figure 1A). We established ten sample sites > 50 m^2^ or 36 m^2^ in Qilian and Maqin counties, known for their dense *Aster* populations; each population site has three replicates across diverse alpine habitats (meadows and mire meadows, Figure 1B). Supporting at least nine dominant species (*Aster flaccidus* (QLAF), *Aster diplostephioides*, etc.), these sites included two distinct *A. diplostephioides* (QLAD and MQAD) populations (Appendix A) and were confirmed to have no history of pesticide use. Surveys combined systematic sampling (15–20-day intervals, timed to phenology, avoiding rain) and targeted monitoring of 30 sites/150 points, focusing on key *Aster* pests. Arthropod sampling at each point used sweep netting (50 nets), pitfall traps (45 total), yellow sticky traps (3/trap), and branch shaking (45 plants). Recorded data included arthropod density (individuals/area/point), developmental stages, and floral damage area. Site layout considered topography to ensure comprehensive sampling across microhabitats and data representativeness for the wider area.

### 2.2. Pest Identification and Molecular Analysis

Collected samples were fixed in 75% ethanol and identified using a combination of morphological (stereomicroscopy, referring to ‘Fauna in China’) and molecular (*COI* gene sequencing) methods. For species not identified morphologically, DNA was extracted (methods adapted from Zhang et al. [24] and Luo et al. [25]), and PCR was amplified using universal *COI* primers (LCO1490/HCO2198) [24]. Sequences were bidirectionally sequenced, and species were confirmed via NCBI BLAST alignment.

### 2.3. Integrated Pest Management Experiments for Major Pests

Integrated pest management experiments were primarily conducted from June to September 2024 at the alpine grassland ecological experimental station in Qilian County (Haibei Prefecture) and the *A. diplostephioides* (QLAD) site in Hailang Village. Prior to experiments, 3000 QLAD plants (~270 pots, 10–12 plants per pot) were transplanted to an open area outside the experimental greenhouse at the Qilian alpine grassland ecological experimental station. Plants were grouped in sets of 6 or 9 per plot for subsequent experiments. One side of the open area was shielded by a ~3 m tall wooden fence to mitigate excessive sunlight and strong winds.

Physical control involved flower head bagging, applied to both potted and field-grown plants. In field plots (QLAD, *Aster souliei* (DRAS), *Aster farreri* (MQAF), *Aster yunnanensis* var. *labrangensis* (MQAY), MQAD), 30 plants were selected per plot (1 plant every 1 m), with three replicates. In potted plants, 24 bagging treatments plants were set up, with 4 plants selected per treatment under each lure condition. Bags fully enclosed the flower heads, with openings tightly sealed, covering the plants from the leaf expansion stage to seed set.

Chemical control included lure-based and chemical agent experiments. Potted QLAD and native plants from field sites were used. Four lure combinations were tested: (1) broad-spectrum fruit fly lure (Ae); (2) Ae + sugar–vinegar solution (Su); (3) Ae + abamectin (Ch); (4) Ae + Su + Ch. Each treatment group included six potted plants, with three lure bottles 1.5 m above the pots. In addition, three sets of clean water control samples were set up and placed separately. There was a total of 15 bait bottles. Lure ratios and concentration gradients followed Appendix A, applied during different growth stages (leaf expansion to full flowering). Field validation was conducted in 5 m × 5 m plots during full flowering. Lures were replaced every 15 days, and records were kept of pest species, abundance, and plant growth.

Chemical agent screening tested five treatments and their combinations: 10% β-Cypermethrin aqueous emulsion + 5% abamectin (DB), 90% crystal dichlorvos (JC), 40% phoxim emulsifiable concentrate (SM), 10% imidacloprid (PA), and 5% broflanilide (SC), along with a water control. Each agent was tested at three concentration gradients (Appendix A), applied during five stages: leaf unfolding stage (LS), initial flowering stage (IS) + LS, peak bloom stage (PS), LS + IS + PS, and IS + PS. Initially, pest counts were recorded before spraying in potted plants to screen for optimal agents, timing, and concentrations. Post-treatment mortality and plant damage rates (PDRs) were assessed at the end of flowering. Potted experiments had three replicates (nine pots per treatment), with counts recorded before and after spraying. Field validation was conducted in 45 experimental plots. Meanwhile, six potted plants treated with water spraying were set up as controls. All chemical preparations and applications followed manufacturer instructions, with plastic boards used to shield non-target plants during spraying. Post-spraying, live and dead pests were counted at 1, 3, and 5 h and 1, 2, 3, and 5 days. Pests and floral tissues were collected, sorted, and stored in cryovials at liquid nitrogen temperatures for subsequent analysis.

### 2.4. Biochemical Index Detection and Data Analysis

Based on potted plant control efficacy, biochemical analysis focused on samples treated with C3 concentration agents (1 h, 5 h, 24 h (D1), and 120 h (D5)). Five resistance-related indices—polyphenol oxidase (PPO), catalase (CAT), phenylalanine ammonia-lyase (PAL), malondialdehyde (MDA) content, and total phenols (TP)—were measured via spectrophotometry [7], while insect acid phosphatase (ACP), alkaline phosphatase (AKP), and glutathione-S-transferase (GST) activities were detected using kit methods. Procedures: 0.1 g of floral tissue per treatment was homogenized in 1 mL of extraction buffer under ice conditions. Supernatants were obtained via centrifugation: ACP (EC 3.1.3.2) and AKP (EC 3.1.3.1) at 4 °C, 12,000 rpm for 15 min; GST (EC 1.8.1.9) at 4 °C, 8000 rpm for 10 min. Enzyme activities were measured via microplate readers following kit and instrument protocols (Suzhou Grace Biotechnology Co., Ltd., Suzhou, China). ACP, AKP, and GST activities were determined at 405, 405, and 340 nm, respectively, with enzyme activity calculated from absorbance changes. Results were expressed as μmol/min/g FW, with three biological replicates per index.

### 2.5. Data Analysis

#### 2.5.1. Calculation Methods for Experimental Indicators

The various indicators involved in this experiment include nine items, including the total number of mature seeds in a single inflorescence, the failure rate, inflorescence fresh weight, total dry weight of mature seeds per inflorescence, 1000-seed weight of mature seeds, plant damage rate (PDR), length of insect tunnels, pest population reduction rate (PRR), and control effect (CE) [26,27,28]. Among them, notable were the percentage of damaged plants (PDR) among every 200 asters in the total number of surveyed plants; PRR (%) = (number of dead pests after treatment/total number of pests before treatment) × 100; CE (%) = [(mortality rate in treated plots − mortality rate in water control)/(100 − mortality rate in water control)] × 100.

#### 2.5.2. Data Statistical Analysis

Data were first tested for normality and homogeneity of variance using Q-Q plots and interquartile range (IQR) tests. ANOVA and Duncan’s multiple comparisons were used to assess differences between groups. Principal component analysis, factor analysis, and heatmap analysis were performed to identify key enzyme systems and their correlations with control efficacy. All statistical analyses were conducted using SPSS v 26.0 (SPSS Inc., Armonk, NY, USA), with tables created in Microsoft Excel 2016 and heatmaps generated via the GenesCloud platform (https://www.genescloud.cn, accessed on 22 May 2025). Other figures were produced using Origin 2021. Data are presented as means ± standard errors.

## 3. Results

### 3.1. Species Diversity and Geographical Distribution Characteristics

A total of 14,678 insect individuals (57 families, 89 genera, 96 species) and 447 spider individuals (7 families, 13 species) were collected from 10 species of *Aster* sample sites in the TRSR (Appendix A). Taxa with substantially higher abundance than others were defined as dominant groups. Distinct differences were observed in dominant insect taxa composition and proportions between regions, as well as among different sampling sites within the same region (Figure 2). In Golog Prefecture’s sites, Diptera (58.89%), Hymenoptera (12.43%), and Hemiptera (9.37%) constituted the dominant orders. Haibei Prefecture exhibited different ordinal dominance: Hemiptera (24.71%), Lepidoptera (22.35%), Coleoptera (16.61%), and Diptera (15.08%) (Figure 2A,B).

At the family level, five sites in Maqin County (MQAF, MQAD, MQAY, MQAA, MQAP) yielded 9652 insect individuals (20 families, 34 genera, 37 species), dominated by Tephritidae (35.72% ± 8.67%). MQAF showed the highest Tephritidae proportion (49.9%), followed by MQAD (46.8%). Dari County sites (DRAS, DRAA) recorded 2373 insects (38 families, 50 genera, 51 species), with Vespidae (26.6%) and Tephritidae (14.2%) dominating DRAS, while Tephritidae (31.9%) and Aphididae (13.3%) prevailed in DRAA. Banma County’s BMAT site exhibited the lowest insect diversity (11 families, 13 genera, 14 species, 173 individuals), dominated by Anthomyiidae (29.5%) and Tephritidae (25.4%). Qilian County sites (QLAD, QLAF) in Haibei contained 2480 insects (51 families, 85 species), with QLAD dominated by Tephritidae (18.2%) and Thripidae (10.4%), versus Thripidae (17.6%) and Miridae (16.4%) in QLAF.

Across most regions (excluding MQAP and MQAA), Lycosidae (45.06% ± 6.98%) and Thomisidae (29.5%) were dominant spider families. Key dominant/persistent species (*Pardosa tesquorum*, *P. astrigera*, *Misumenops tricuspidatus*, *Xysticus striatipes*) exhibited significant regional preferences (Table 1 and Appendix A). The core pest *T. angustipennis* caused severe damage to aster inflorescences, and feeding caused the stems of the asters to become hollow, the flower cups to be damaged, and many plants to become hollow and even rot away. Golog Prefecture additionally had *Rhopalomyia giraldii* and *Melitaea cinxia*, whereas Haibei’s supplementary dominant species were *Frankliniella intonsa* and *Adelphocoris lineolatus* (Appendix A, Appendix A).

### 3.2. Spatiotemporal Dynamics of Dominant Species

#### 3.2.1. Spatiotemporal Dynamics of Insect Dominants

Across sampling sites, the abundance of dominant insect species exhibited significant temporal fluctuations. Furthermore, within the same sampling sites, there were significant differences in both types and quantities of the various species present (*p* < 0.05, Appendix A, Appendix A). In Golog Prefecture, the population size of *T. angustipennis* (Ta) displayed interannual consistency across eight sites, peaking in mid–late August (*p* < 0.05). Notably, MQAF, MQAD, and MQAY consistently harboured higher Ta densities than other sites (*p* < 0.05), suggesting a positive correlation with local temperatures. Mid-August comparisons revealed significantly higher Tephritidae proportions in Maqin County plots compared to Dari County (*p* < 0.001) (Figure 3A,B and Appendix A), highlighting altitudinal gradient effects on pest distribution (Figure 1A). *R. giraldii* (Rg) exhibited spatial heterogeneity in its phenology, peaking in mid-July (MQAY, DRAA, BMAT), early–mid-August (MQAF, MQAA, DRAA), and late August (MQAD), synchronized with the bud of *Aster* spp. differentiation. *Hylemyza partita* (Hp) peaked in early August at MQAY and MQAP but showed progressive increases in six other sites, with significant late-August surges (*p* < 0.05). BMAT maintained consistently higher Hp densities than other dominants (*p* < 0.05). *M. cinxia* (Mc) displayed gradual increases in Golog, peaking in September and differing significantly from earlier pest phases (*p* < 0.05). *Campiglossa loewiana* (Cl) drove structural community changes at MQAY in August (*p* < 0.05), while altitudinal gradients significantly influenced Tephritidae proportions (*p* < 0.001, Table 2, Appendix A).

In Haibei Prefecture, Ta exhibited unimodal dynamics, peaking in mid–late August with QLAD densities significantly exceeding QLAF (*p* < 0.05). *F. intonsa* (Fi) peaked mid-August in both sites, reflecting microhabitat selection preferences. Mc showed continuous increases in QLAD until late September, significantly surpassing QLAF (*p* < 0.05), indicative of host resource allocation divergences (Figure 3C,D). Miridae pests exhibited temporal lags: *A. lineolatus* (Ai) peaked in late July (QLAD) versus mid-August (QLAF), while *Apolygus lucorum* (Al) declined sharply after mid–late July peaks (*p* < 0.05). *Curculionidae* sp. (Cs) peaked in early August before a significant decline (*p* < 0.05), with higher QLAD densities than QLAF (Table 3, Appendix A). Overall, both regions exhibited unimodal abundance patterns, with Haibei peaks occurring from mid-July to August and Golog peaks concentrated in August. Significant inter-site variations were observed (QLAD > QLAF; MQAY/MQAD/MQAF > other sites). Spatiotemporal distribution divergences (*p* < 0.05) occurred between two key pests (Ta and Mc), with MQAF, MQAY, and MQAD identified as core outbreak zones due to sustained density increases. These patterns aligned closely with host plant phenology, reflecting regional divergence in occurrence modes driven by differences in host phenophases (Figure 3, Appendix A).

#### 3.2.2. Spatiotemporal Dynamics of Spider Dominants

Dominant spider species abundance exhibited temporal fluctuations across sampling sites with distinct peak periods (Appendix A). *P. tesquorum* displayed an initial increase followed by a decline in DRAS and QLAA sites, peaking in early August. *P. astrigera* populations gradually decreased across all sites, reaching maximum abundance in mid–late June before subsequent reductions. *M. tricuspidatus* exhibited unimodal dynamics, peaking in mid-July. *X. striatipes* demonstrated an increase–decline pattern, peaking in early August at MQAF and DRAA with marked post-peak reductions, whereas in DRAS, its peak occurred in mid-July, mirroring the temporal trends of dominant pest species. Additionally, a regional analysis showed that Lycosidae spiders (*P*. *tesquorum*) reached their peak abundance in early August at the DRAS site, negatively correlating with tephritid larval density (*p* < 0.05). Thomisidae spiders (*X. striatipes*) at the MQAF site peaked in mid-July, synchronized with aphid population surges. The QLAD site was dominated by Lycosidae, while QLAF favoured Thomisidae, reflecting spatiotemporal heterogeneity in predatory spider distributions and underscoring microhabitat influences on natural enemy assemblages.

### 3.3. Comparison of Physical and Chemical Control Methods

Bagging significantly reduced the PDR (*p* < 0.05), but it led to an increased rate of seed sterility (*p* < 0.01), as well as significantly decreasing the weight of individual inflorescences, the total number of mature seeds per inflorescence, and the total dry weight of mature seeds per inflorescence (*p* < 0.05) (Appendix A). Lures and their combinations showed a significant attraction effect on fruit flies infesting potted *Aster* plants (*p* < 0.05). Additionally, the lure combination treatment Ae + Su + Ch achieved the highest mortality efficiency against *T. angustipennis* adults, significantly surpassing single lure applications, and was most effective when applied during the IS, full flowering stage, and fruit development stage (*p* < 0.05). This finding was also validated in field lure experiments (Table 4).

Results from the potted chemical control experiments indicated that pesticide type (DB, JC, SM, PA, SC), application timing (S), and concentration (C) all significantly influenced the PDR, PRR, and CE (*p* < 0.05, Appendix A). Multi-stage application (spraying at the LS, IS, and PS) significantly enhanced control efficacy, with the LS + IS + PS three-application scheme demonstrating the optimal effect (Appendix A). The combination of β-Cypermethrin + avermectin (DB) showed significantly superior control efficacy against *T. angustipennis* compared to other combinations (Figure 4A), with higher concentrations (C3) yielding better results (C3 > C2 > C1) (Figure 4B). Analysis of application timing revealed that the initial application at the LS plays a critical role in disrupting the IS of the pest (Figure 4C).

Field experiment results showed that pesticide type (T) and concentration (C) significantly influenced both the PRR and CE (*p* < 0.05, Appendix A). The DB combination applied at both the LS + PS exhibited the highest efficacy (Figure 4D), resulting in a significant drop in pest density within 5 h of application, although the efficacy duration was relatively short, but this achieves an 80% reduction in pest population within 24 h (Appendix A). SM was less effective. SC had a slower onset (48–72 h), but its control effect against lepidopteran larvae (e.g., *M. cinxia*) was prolonged (7 days), and it could simultaneously control both fruit fly and lepidopteran complex damage. Regarding application concentration, the C3 concentration of all pesticides showed significantly superior field control efficacy compared to C2 and C1 (Appendix A, Figure 4E,F), consistent with the results from the potted experiments.

### 3.4. Enzyme Activity Responses and Mechanistic Analysis

#### 3.4.1. Plant Responses

Pesticide treatments significantly affected floral receptacle enzyme activity and secondary metabolite levels (*p* < 0.05). Within 24 h, all CAT, PPO, and PAL activities were boosted, plus MDA and TP content (Figure 5A–E). DB and JC treatments strongly enhanced antioxidant (CAT) and phenylpropanoid (PAL, PPO) pathways, boosting metabolite (e.g., TP) accumulation and defence. SC and SM treatments caused transient, specific activity peaks (e.g., 48 h) with generally lower intensity than DB and JC (*p* < 0.05). MDA elevation at 5 h, followed by fluctuations, signified persistent oxidative stress. Enzyme activities and metabolite levels in all treatment groups remained significantly higher than in the water control (CK), confirming systemic activation of plant defence mechanisms under chemical stress.

#### 3.4.2. Pest Detoxification Responses

*T. angustipennis* detoxification enzyme activities were significantly modulated by pesticide type (*p* < 0.05). DB and JC treatments induced rapid ACP activity peaks (2.16 and 2.05 μmol/min/g, respectively) within 5 h, significantly earlier than the PA, SM, and SC groups (peak delays to 24 h; Figure 5F). GST activity under DB treatment peaked at 5 h (2.21 μmol/min/g), doubling CK levels (Figure 5G). AKP activity synchronously peaked at 24 h across all treatments (1.09–2.21 μmol/min/g) before sharp declines (Figure 5H). Notably, the DB and JC groups uniquely sustained higher detoxification enzyme activities than other treatments during most timepoints (*p* < 0.01), reflecting both rapid inhibition and strong induction that may foster pest resistance evolution.

#### 3.4.3. Dominant Factor Analysis

PCA and factor analyses examined *Aster* spp. resistance indices, pest detoxification enzymes, and control efficacy post-treatment. The first two PCA axes accounted for 53.2% of variance (Figure 6A). Resistance index clustering confirmed significant, treatment-specific plant–pest response divergence. Significant positive correlations emerged between control efficacy and CAT, PAL, AKP, ACP, and GST activities, whereas MDA content negatively correlated with efficacy, highlighting their critical role in pest reduction (Figure 6A,B). Plant CAT and pest GST activities were the primary drivers of efficacy variation (cumulative explanation rates 94.26% and 97.53%), with MDA inversely linked to control. This underscores plant–pest interactions determining efficacy divergence. Interactive heatmaps further identified CAT and GST as key factors shaping efficacy differences (Figure 7).

## 4. Discussion

### 4.1. Optimization and Limitations of Sampling and Identification Methods

This study employed a combination of trapping and sweep netting to survey arthropods associated with *Aster* spp. habitats. Our findings underscore the importance of methodological selection based on target taxa and behaviour, revealing inherent biases in collection efficiency. Web-based methods proved effective for capturing web-building spiders, while pitfall traps yielded higher numbers of ground-dwelling species [25], highlighting the need for tailored approaches reflecting the behavioural ecology of spiders in future research. Integrating morphological identification with *COI* gene barcoding significantly enhanced taxonomic precision, resolving species identities for 121 specimens that eluded morphological determination [29,30]. This reinforces the necessity of molecular tools for accurately assessing high-altitude arthropod diversity. However, the encountered limitations, such as *COI* amplification failures in some samples, reflect the primer-specific constraints common in molecular studies [22,31]. This necessitates the exploration of alternative nuclear markers (e.g., *ITS* or *28S* rDNA) for improved phylogenetic resolution, particularly in taxonomically complex groups.

### 4.2. Structural and Dynamic Drivers of Alpine Insect Communities

Our survey recorded 96 insect species, with markedly higher diversity in Golog Prefecture compared to Haibei, demonstrating the strong influence of regional environmental heterogeneity and habitat variation [25]. Diptera (Tephritidae) and Hemiptera (Miridae, Cecidomyiidae) were the dominant groups across sites, exhibiting summer-unimodal population dynamics peaking in mid–late August. This pattern correlates with general alpine insect activity peaks and the phenology of *Aster* flowering [22,32,33]. The peak of the key pest Tephritidae species, *T. angustipennis*, in August aligns with regional seasonal trends but contrasts with July peaks observed in the Sichuan Basin [34], suggesting that altitudinal gradients modulate its phenology through differences in accumulated temperature and host plant synchrony [35]. Furthermore, insect community dynamics resulted from complex, multifactorial interactions involving abiotic drivers (e.g., temperature and precipitation extremes in July–August) [36], biotic factors (e.g., *Aster* resources), and anthropogenic impacts (e.g., grazing disturbances) [37]. We observed that high vegetation covers generally supported greater insect diversity, but the grazing pressure may alter the plant community structure, potentially increasing the availability of alternative host plants for Miridae or creating microclimates conducive to their development, illustrating how niche differentiation and habitat heterogeneity shape these patterns [16,38]. Notably, Tephritidae species likely play dual roles as both pests (damaging *Aster*) and pollinators, necessitating management strategies that balance control with ecological function [39,40,41]. Collectively, these findings provide critical insights for predictive modelling of alpine pest outbreaks and inform the development of IPM approaches suitable for high-altitude ecosystems.

### 4.3. Efficacy and Ecological Trade-Offs of Physical–Chemical Integrated Control

Effective pest management is vital for grassland health and ecosystem stability [19], particularly for *Aster* cultivation, where *T. angustipennis* and *M. cinxia* are primary culprits. While flower head bagging effectively isolates pests, it negatively impacts pollination, increasing seed sterility—a clear trade-off highlighting the ecological costs of physical control [39]. Thus, its use is recommended only sparingly in focal areas. In contrast, integrated chemical strategies employing specific attractants for fruit flies combined with low-toxicity pesticides (e.g., abamectin) offered effective control while minimizing chemical input, especially during aster’s critical growth stages [42]. This approach aligns with strategies used for other pests of Tephritidae [14,20], demonstrating that attractant-based chemical applications during flowering can reduce populations with less environmental impact, consistent with sustainable agriculture principles [14,43]. Among tested chemical combinations, 10% β-Cypermethrin + 5% avermectin (DB) provided rapid control of *T. angustipennis* (>85% mortality within 24 h), yet its short persistence (declining after 5 days) and induction of detoxification enzymes (GST, ACP, AKP) necessitate rotational use [44]. Conversely, 5% broflanilide (SC) offered persistent control against lepidopteran larvae, suggesting suitability for long-term prevention [45]. These results confirm that strategic application timing, matched to the *Aster* growth cycle (LS, IS, PS), is crucial for efficacy [46]. While chemical control is effective, the potential impact on non-target organisms, particularly pollinators, requires careful long-term monitoring to ensure ecological safety [47].

### 4.4. Biochemical Regulatory Networks and Key Factors in Plant–Insect Interaction

The interaction between chemical control agents and the biochemical defences of plants and insects formed a complex regulatory network. Application of chemical agents triggered defence responses in *Aster* (e.g., activating enzymes like CAT, PPO, and PAL and secondary metabolites like TP), while insects countered these effects primarily through detoxification enzymes like AKP and GST (peaking at 2.21 μmol/min/g) [48,49,50]. PCA and interaction network analysis identified plant CAT activity and insect GST activity as the key factors driving differences in control efficacy. High plant CAT activity enhances oxidative stress tolerance but may also deter insects via the H_2_O_2_ signalling pathway [48,49]. Conversely, high insect GST activity confers resistance by metabolically detoxifying the applied chemicals, creating an antagonistic cycle of “plant defence–insect anti-defence” [50]. This interplay suggests that future strategies could involve developing GST inhibitors to synergize with chemical agents [51,52] or priming plant defence pathways (e.g., phenylpropanoid pathway using methyl jasmonate) to enhance control timeliness [53]. Therefore, dynamic monitoring focused on key enzyme activity thresholds is essential for optimizing application timing towards a “need-based control” approach [54].

### 4.5. Necessity for Implementing Integrated Pest Management and a Sustainable Pest Management Framework for Insects in High-Altitude Ecosystems

The success of IPM fundamentally relies on real-time, precise monitoring [55]. Our findings, resonating with studies on *Bactrocera dorsalis* [44] and *Spodoptera litura* [42], confirm that continuous monitoring is essential for effective control, especially given the influence of climate change and agricultural activities on the spatiotemporal dynamics of key pests like *T. angustipennis* [10]. This necessitates the establishment of long-term monitoring networks specifically for *Aster* pests. Furthermore, dynamic population changes dictate the optimal timing and specificity of control measures [44,56], requiring adaptive strategies that refine interventions based on regional insect patterns, weather forecasts, and *Aster* phenology [47].

Building on these insights, this study proposes a structured “Monitoring–Warning–Intervention” tripartite IPM system [57]: (i) a monitoring layer utilizing *COI* barcoding and remote sensing for predictive models, dynamically calibrated with altitude and phenology data [58]; (ii) a warning layer that anticipates resistance risks using enzyme activity thresholds (e.g., GST > 1.8 μmol/min/g) to enable proactive planning [59]; and (iii) an intervention layer employing targeted controls using attractants and selective pesticides during key phenological stages (IS + PS), supplemented by ecological enhancements like planting companion plants to bolster natural enemy populations (ichneumonids, ladybugs, spiders) [40]. This framework balances immediate pest suppression with long-term resistance management through multi-targeted actions (physical isolation, chemical regulation, ecological optimization), offering a sustainable paradigm for high-altitude medicinal plant cultivation [59]. Future research must prioritize screening additional environmentally friendly pesticides and evaluating their compatibility with physical control methods [43]. By integrating robust monitoring, green control options, targeted chemical applications, and ecological regulation, a resilient IPM system can be established to ensure the sustainable utilization of *Aster* species in high-altitude regions [60].

## 5. Conclusions

*Aster* spp., serving as pivotal species for ecological restoration in the degraded alpine grasslands of the TP, face significant challenges from insect infestations that jeopardize both germplasm conservation and ecosystem recovery. This study, across 10 Qinghai TRSR sampling sites, identified 109 arthropod species, dominated by Diptera (Tephritidae) and Hemiptera (Miridae), with Tephritis angustipennis being the primary pest. Its population density peaked in mid-to-late August (*p* < 0.05) and exhibited a significant tracking relationship with predatory spiders (Lycosidae and Thomisidae), emphasizing the necessity for intensified monitoring and timely intervention during this critical period. Integrating spiders and other predatory natural enemies into pest management systems is strongly recommended.

This study suggests an IPM strategy, encompassing the following aspects: (i) Physical barriers: flower cluster bagging combined with a composite attractant (broad-spectrum fruit fly + sugar–vinegar) significantly reduces pest incidence and improves adult capture rates. (ii) Precision application program: targeted application of a composite insecticide (DB: 10% β-Cypermethrin aqueous emulsion (9 mL/acre) + 5% abamectin microemulsion (20 mL/acre)) during LS and IS achieves an 80% pest reduction within 24 h. Adding 5% broflanilide during full flowering extends efficacy to 7 days. (iii) Natural enemy: Lycosidae and Thomisidae spiders demonstrated substantial potential for natural pest population regulation. Mechanistic analysis indicates that divergent plant CAT and pest GST activities drive control efficacy (cumulative explanatory power reached 94%). The developed “four-in-one” IPM framework—integrating physical exclusion, precision trapping, targeted chemical application, and natural enemy synergy—can significantly enhance pest control efficiency, providing a robust theoretical and practical basis for restoring degraded alpine grasslands and ensuring sustainable medicinal plant management in high-altitude ecosystems.

## Figures and Tables

**Figure 1 insects-16-00861-f001:**
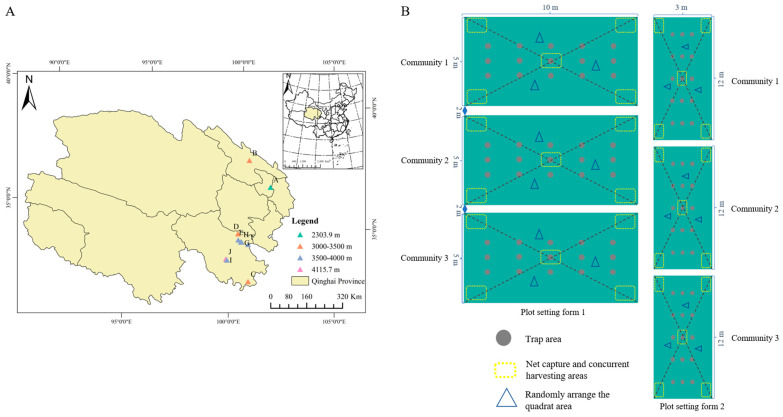
Distribution of sampling points in different aster sample sites (**A**) and map of sample plot settings (**B**).

**Figure 2 insects-16-00861-f002:**
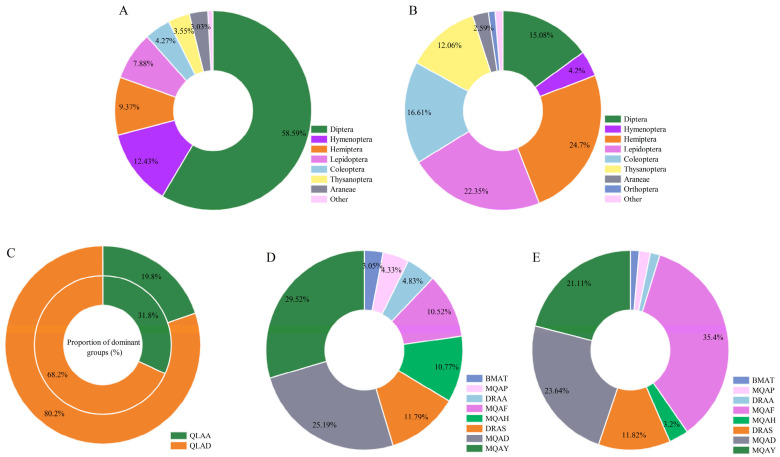
Percentage of dominant groups in different sampling sites. Note: (**A**,**B**) distributions represent the proportion of dominant pest categories of each order in Golog Tibetan Autonomous Prefecture and Haibei Tibetan Autonomous Prefecture, respectively; and other represents taxa with a proportion of less than 1%. (**C**) is a comparison of the number of Diptera (outer circle) and Hemiptera (inner circle) within the two sampling sites in Haibei Tibetan Autonomous Prefecture, and (**D**,**E**) is the proportion of insects in the dominant taxa of Hemiptera and Diptera in Golog Tibetan Autonomous Prefecture, respectively, in the respective sampling sites, and the labels with proportions of less than 2% are hidden.

**Figure 3 insects-16-00861-f003:**
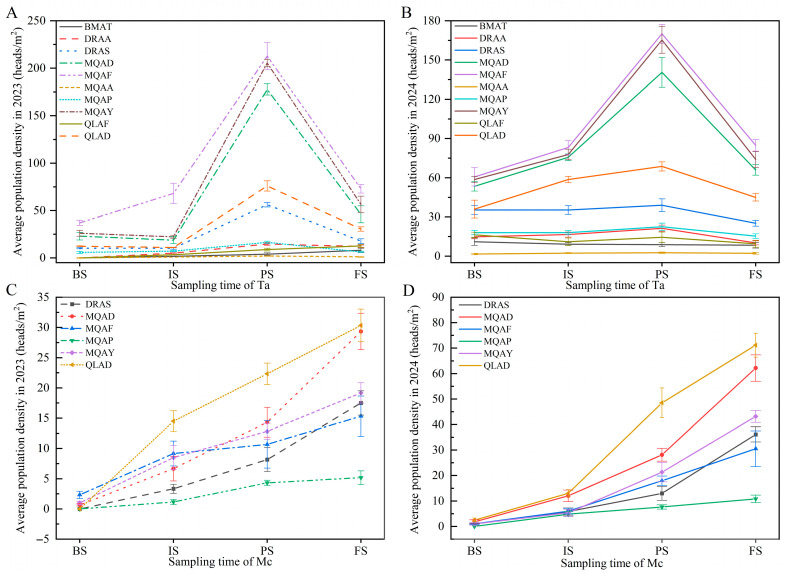
Dynamic analysis of *T. angustipennis* (Ta) and *M. cinxia* (Mc). Note: (**A**,**B**) illustrate the average population density of Ta across different sites in 2023 and 2024, respectively. Similarly, (**C**,**D**) depict the average population density of Mc in various sites during the years 2023 and 2024. The population size of the Ta and Mc in the LS is not shown in the figure because during LS in either year, no such pests were detected in any of the experimental fields.

**Figure 4 insects-16-00861-f004:**
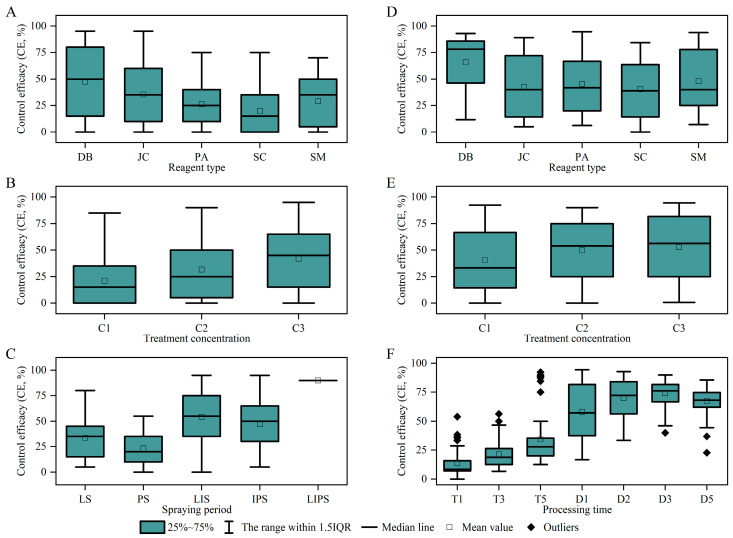
Comparison of prevention and treatment effects under different treatments. Note: left (**A**–**C**) is the result of a pot experiment, and right (**D**–**F**) is the result of field experiment.

**Figure 5 insects-16-00861-f005:**
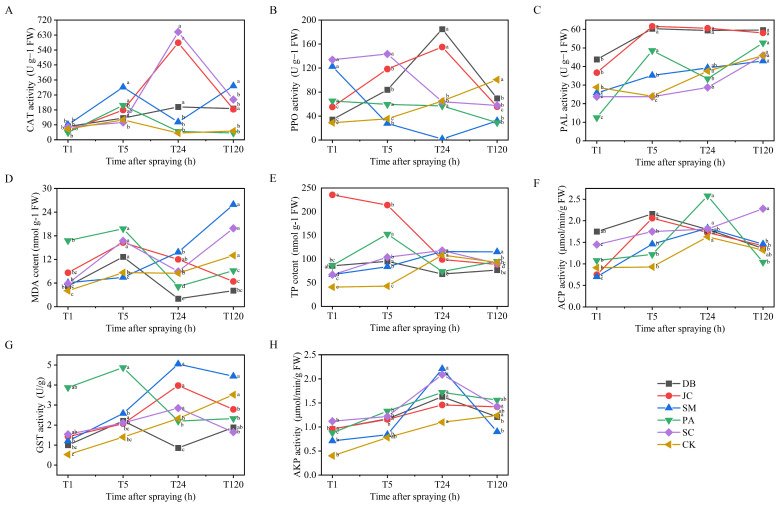
Changes in the resistance enzyme activity of plant and the detoxifying enzyme activity of *T. angustipennis* larvae after pesticide treatment. Note: (**A**–**E**) show the comparison of changes in the resistance enzyme systems of QLAD flowers under different pesticide treatments; (**F**–**H**) show the comparison of changes in the detoxifying enzyme activity in the bodies of *T. angustipennis* larvae. The lowercase letters in the figure represent the significant differences in enzyme activities or metabolite contents of different plants or insects at different times after spraying. Different lowercase letters indicate significant differences (*p* < 0.05), while the same lowercase letters indicate no significant differences (*p* > 0.05).

**Figure 6 insects-16-00861-f006:**
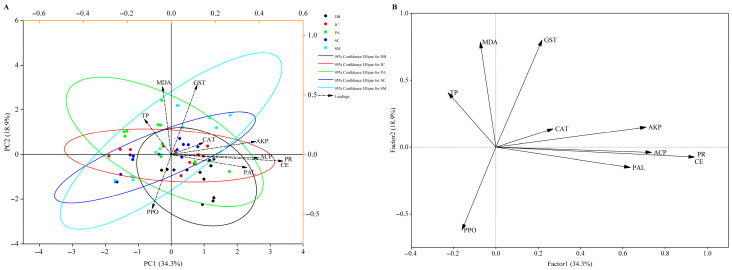
PCA (**A**) and factor (**B**) analysis of plant and pest resistance indicators and control effectiveness under different treatment agents.

**Figure 7 insects-16-00861-f007:**
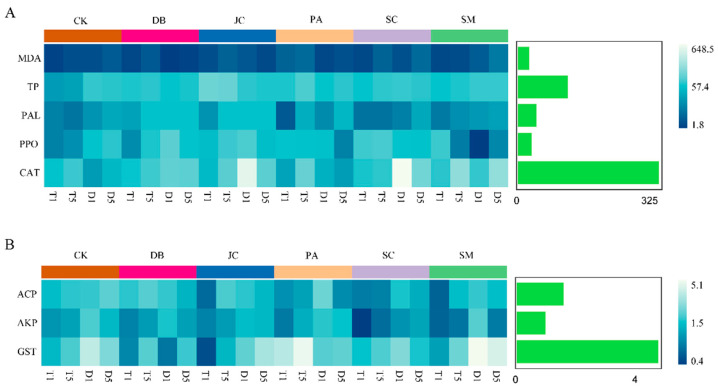
Heatmap showing changes in the enzyme activity at different reaction times after treatment with various agents. (**A**) Comparison of changes in the resistance enzyme systems of QLAD flowers under different pesticide treatments; (**B**) comparison of changes in the detoxifying enzyme activity in the bodies of *T. angustipennis* larvae. Note: The bar chart shows the importance score of these significantly different substances in each comparison group. The longer the bar, the greater the importance score of enzyme activity or secondary metabolite content to the groups, indicating that the substance is more important to the samples between groups. Based on this, it can be considered that the substances with larger importance scores are the marker species of the differences between groups.

**Table 1 insects-16-00861-t001:** Statistics of the total sample volume at each sampling site in 2023.

Sampling Sites	Growth Period of Golog Tibetan Autonomous Prefecture					Total Number of Samples	Sample Proportion
	LS	BS	IS	PS	FS		
MQAF	286	837	1069	1399	292	3883	30.87%
MQAD	254	449	654	987	165	2509	19.95%
MQAY	211	571	621	862	204	2469	19.63%
MQAA	84	147	159	145	56	591	4.70%
MQAP	15	35	54	111	21	236	1.88%
BMAT	21	59	69	23	15	187	1.49%
DRAS	281	501	667	757	151	2357	18.74%
DRAA	57	67	85	89	49	347	2.76%
Total	1209	2666	3378	4373	953	12,579	100.00%
Sampling sites	Growth period of Haibei Tibetan Autonomous Prefecture					Total Number of Samples	Sample Proportion
	LS	BS	IS	PS	FS		
QLAD	125	322	511	658	225	1841	72.31%
QLAF	65	91	124	356	69	705	27.69%
Total	190	413	635	1014	294	2546	100.00%

Note: These sampling times are closely related to the growth period of *Aster* spp. That is, stage dates for Golog/Haibei (LS to FS dates) were leaf unfolding (6.20–6.25/6.15–6.17), budding (7.16–7.20/7.22–7.24), initial flowering (8.5–8.12/8.14–8.16), peak bloom (8.25–8.30/8.20–8.22), and fruiting (9.15–9.20/9.10–9.12). The same staging applies below.

**Table 2 insects-16-00861-t002:** Differential analysis of the number of dominant species of insects in each sampling site in Golog Tibetan Autonomous Prefecture.

Sampling Sites	Dominant Species Name (Abbreviation)	Growth Period				
		LS	BS	IS	PS	FS
MQAF	*Tephritis angustipennis* (Ta)	0	56 ± 0.33 B	122 ± 0.33 A	195 ± 1.2 A	109 ± 4.48 A
	*Campiglossa loewiana* (Cl)	0	58 ± 0.36 A	33 ± 0.67 C	56 ± 0.33 B	46 ± 0.67 B
	*Hylemyza partita* (Hp)	8 ± 0.33 A	13 ± 0.19 D	36 ± 0.58 B	38 ± 0.33 C	31 ± 0.67 C
	*Rhopalomyia giraldii* (Rg)	0	20 ± 0.33 C	32 ± 1.67 C	22 ± 0.88 D	8 ± 0.35 F
	*Parasitoid wasp* (Pw)	3 ± 0.67 B	9 ± 0.33 E	16 ± 1.12 D	23 ± 1.15 D	19 ± 0.33 E
	*Aproaerema anthyllidella* (Aa)	0	4 ± 0.23 F	12 ± 0.88 E	23 ± 1.53 D	27 ± 1.67 CD
	*Melitaea cinxia* (Mc)	0	1 ± 0.58 G	13 ± 1.2 E	16 ± 0.67 E	23 ± 0.67 DE
MQAD	*Tephritis angustipennis* (Ta)	0	47 ± 1.12 A	62 ± 0.33 A	99 ± 1.2 A	81 ± 0.58 A
	*Campiglossa loewiana* (Cl)	0	8 ± 0.33 C	15 ± 0.23 E	46 ± 0.32 B	29 ± 0.58 B
	*Hylemyza partita* (Hp)	0	6 ± 0.33 E	25 ± 1.53 C	33 ± 0.67 D	19 ± 0.58 D
	*Rhopalomyia giraldii* (Rg)	0	7 ± 0.19 D	27 ± 0.33 B	35 ± 0.33 C	23 ± 1.23 C
	*Parasitoid wasp* (Pw)	4 ± 0.67 A	12 ± 0.33 B	18 ± 0.88 D	6 ± 0.28 F	0
	*Aproaerema anthyllidella* (Aa)	0	0	7 ± 0.33 F	9 ± 0.33 E	13 ± 1 E
	*Melitaea cinxia* (Mc)	0	0	1 ± 0.33 G	1 ± 0.05 G	2 ± 0.26 F
MQAY	*Tephritis angustipennis* (Ta)	0	35 ± 0.67 A	46 ± 0.67 A	109 ± 0.58 A	53 ± 0.58 A
	*Campiglossa loewiana* (Cl)	7 ± 0.58 B	10 ± 0.33 C	29 ± 0.58 B	21 ± 0.33 B	2 ± 0.25 B
	*Hylemyza partita* (Hp)	4 ± 0.33 C	11 ± 0.33 C	16 ± 1.21 D	9 ± 0.27 D	3 ± 0.36 B
	*Rhopalomyia giraldii* (Rg)	7 ± 0.05 B	30 ± 0.88 B	23 ± 0.59 C	19 ± 0.88 C	4 ± 0.28 B
	*Parasitoid wasp* (Pw)	4 ± 0.67 C	5 ± 0.58 D	6 ± 0.33 E	1 ± 0.58 F	5 ± 1.2 B
	*Aproaerema anthyllidella* (Aa)	1 ± 0.33 D	2 ± 0.33 E	2 ± 0.33 F	3 ± 0.33 E	6 ± 0.39 B
	*Melitaea cinxia* (Mc)	0	1 ± 0.33 E	1 ± 0.33 F	3 ± 0.01 E	7 ± 0.47 B
MQAA	*Tephritis angustipennis* (Ta)	12 ± 7.25 A	21 ± 8.01 A	42 ± 2.58 A	30 ± 7.67 A	9 ± 2.59 A
	Thysanoptera sp. (Ts)	0	9 ± 0.33 AB	18 ± 0.58 AB	30 ± 0.33 A	12 ± 0.33 A
	*Rhopalomyia giraldii* (Rg)	4 ± 0.67 A	7 ± 0.58 B	12 ± 0.67 AB	2 ± 0.33 B	0
	*Acyrthosiphon pisum* (Ap)	0	2 ± 0.33 B	4 ± 0.29 B	1 ± 0.67 B	1 ± 0.33 B
	*Aphis craccivora* (Ac)	0	1 ± 0.33 B	2 ± 0.33 B	4 ± 0.33 B	1 ± 0.25 B
	*Apolygus lucorum* (Al)	1 ± 0.67 A	2 ± 0.33 B	2 ± 0.67 B	1 ± 0.33 B	0
	*Hylemyza partita* (Hp)	0	1 ± 0.33 B	1 ± 0.58 B	2 ± 0.58 B	1 ± 0.33 B
Sampling sites	Dominant species name (abbreviation)	Growth period				
		LS	BS	IS	PS	FS
MQAP	*Tephritis angustipennis* (Ta)	0	2 ± 0.33 A	4 ± 0.33 AB	6 ± 0.33 A	4 ± 0.33 A
	*Apolygus lucorum* (Al)	0	2 ± 0.88 A	5 ± 0.33 A	2 ± 0.33 B	1 ± 0.58 DE
	*Campiglossa loewiana* (Cl)	0	0	2 ± 0.33 DE	5 ± 0.58 A	2 ± 0.33 BC
	*Tephritis femoralis* (Tf)	0	1 ± 0.33 AB	3 ± 0.33 BC	3 ± 0.58 B	1 ± 0.67 CD
	*Melitaea cinxia* (Mc)	0	0	1 ± 0.58 E	2 ± 0.33 B	3 ± 0.58 AB
	*Acyrthosiphon pisum* (Ap)	1 ± 0.67 A	2 ± 0.33 A	2 ± 0.33 CD	0	0
	*Hylemyza partita* (Hp)	0	1 ± 0.29 AB	2 ± 0.33 CD	2 ± 0.63 B	0
DRAS	*Tephritis angustipennis* (Ta)	0	5 ± 0.33 C	30 ± 0.33 C	38 ± 0.33 B	16 ± 0.33 B
	*Euodynerus dantici* (Ed)	7 ± 0.58 A	18 ± 0.58 A	56 ± 1.2 A	71 ± 0.88 A	31 ± 0.58 A
	*Rhopalomyia giraldii* (Rg)	0	8 ± 0.58 B	44 ± 0.33 B	29 ± 0.47 C	14 ± 0.33 B
	*Symphoromyia crassicornis* (Sc)	2 ± 0.33 B	7 ± 0.58 B	12 ± 0.33 D	29 ± 0.33 C	0
	*Philonthus nitidus* (Pn)	0	2 ± 0.33 D	8 ± 0.33 DE	8 ± 0.33 E	15 ± 0.35 B
	*Hylemyza partita* (Hp)	0	2 ± 0.33 D	6 ± 0.58 E	12 ± 0.33 D	3 ± 0.33 C
	*Lygus pratensis* (Lp)	1 ± 0.33 C	3 ± 0.33 D	12 ± 0.33 D	5 ± 0.01 F	3 ± 0.33 C
DRAA	*Tephritis angustipennis* (Ta)	0	5 ± 0 A	12 ± 0.58 A	7 ± 0.87 A	4 ± 0.58 A
	*Tephritis femoralis* (Tf)	1 ± 0.33 BC	2 ± 0.67 BC	4 ± 0.33 B	2 ± 0.33 C	1 ± 0.05 C
	Geotrupidae sp. (Gs)	2 ± 0.33 A	3 ± 0.33 B	4 ± 0.58 B	0	0
	*Aphis craccivora* (Ac)	1 ± 0.33 BC	1 ± 0.33 C	2 ± 0.67 C	3 ± 0.33 B	2 ± 0.33 B
	Curculionidae sp. (Cs)	0	1 ± 0.33 C	5 ± 0.67 B	1 ± 0.33 D	0
	*Rhopalomyia giraldii* (Rg)	1 ± 0.33 B	2 ± 0.33 BC	2 ± 0.33 C	1 ± 0.33 D	0
	*Acyrthosiphon pisum* (Ap)	0	1 ± 0.41 C	1 ± 0.48 C	3 ± 0.29 CD	1 ± 0.29 CD
BMAT	*Tephritis angustipennis* (Ta)	0	2 ± 0.33 AB	3 ± 0.58 A	5 ± 0.67 A	1 ± 0.33 B
	*Hylemyza partita* (Hp)	2 ± 0.33 A	3 ± 0.67 A	3 ± 0.33 A	6 ± 0.88 A	5 ± 0.67 A
	*Acyrthosiphon pisum* (Ap)	1 ± 0.29 B	2 ± 0.33 A	3 ± 0.88 A	1 ± 0.33 B	1 ± 0.33 B
	Mantodea sp. (Ms)	1 ± 0.05 B	2 ± 0.33 AB	3 ± 0.58 A	1 ± 0.33 B	1 ± 0.33 B
	*Rhopalomyia giraldii* (Rg)	0	2 ± 0.33 AB	2 ± 0.33 AB	0	0
	*Campiglossa loewiana* (Cl)	0	1 ± 0.33 B	1 ± 0.33 B	2 ± 0.33 B	1 ± 0.33 B
	*Aphis craccivora* (Ac)	1 ± 0.29 BC	1 ± 0.48 B	1 ± 0.48 B	1 ± 0.75 B	0

Note: In the table, uppercase letters indicate differences in longitudinal comparisons, which compare the number of different insect species at the same time point; identical letters indicate no significant difference (*p* > 0.05), while different letters signify a statistically significant difference (*p* < 0.05).

**Table 3 insects-16-00861-t003:** Differential analysis of the number of dominant species of insects in the two sampling sites in Haibei Tibetan Autonomous Prefecture.

Sampling Sites	Dominant Species Name (Abbreviation)	Growth Period				
		LS	BS	IS	PS	FS
QLAD	*Melitaea cinxia* (Mc)	0	0	9 ± 0.33 D	15 ± 0.33 C	19 ± 0.33 A
	*Frankliniella intonsa* (Fi)	2 ± 0.33 C	6 ± 0.33 DE	25 ± 0.33 B	19 ± 0.67 B	12 ± 0.33 C
	*Adonia variegata* (Av)	2 ± 0.33 C	4 ± 0.33 F	2 ± 0.58 G	1 ± 0.33 F	0
	*Acyrthosiphon pisum* (Ap)	5 ± 0.58 AB	7 ± 0.58 D	13 ± 0.67 C	3 ± 0.58 DE	2 ± 0.33 D
	*Adelphocoris lineolatus* (Ai)	6 ± 0.58 A	11 ± 0.33 B	7 ± 0.58 E	3 ± 0.67 E	2 ± 0.33 D
	Curculionidae sp. (Cs)	2 ± 0.33 C	6 ± 0.33 E	9 ± 0.33 D	4 ± 0.33 D	0
	*Apolygus lucorum* (Al)	4 ± 0.58 B	9 ± 0.33 C	4 ± 0.33 F	3 ± 0.33 DE	0
	*Tephritis angustipennis* (Ta)	0	19 ± 0.33 A	33 ± 0.67 A	41 ± 0.33 A	16 ± 0.33 B
QLAF	*Gynaephora menyuanensis* (Gm)	5 ± 0.58 A	9 ± 0.33 A	13 ± 0.58 A	3 ± 0.33 B	0
	*Frankliniella intonsa* (Fi)	4 ± 0.33 B	7 ± 0.33 A	10 ± 0.58 B	5 ± 0.33 A	0
	*Adelphocoris lineolatus* (Ai)	3 ± 0.33 C	9 ± 0.33 A	13 ± 0.67 A	1 ± 0.33 C	0
	*Rhopalomyia giraldii* (Rg)	0	5 ± 0.33 B	7 ± 0.33 C	1 ± 0.33 C	0
	Curculionidae sp. (Cs)	1 ± 0.33 D	2 ± 1.2 DE	3 ± 0.33 DE	3 ± 0.67 B	1 ± 0.33 A
	*Apolygus lucorum* (Al)	3 ± 0.58 C	4 ± 0.67 BC	2 ± 0.33 E	0	0
	*Hylemyza partita* (Hp)	1 ± 0.33 DE	3 ± 0.58 CD	4 ± 0.33 D	0	0
	*Tephritis angustipennis* (Ta)	0	1 ± 0.67 E	2 ± 0.33 E	3 ± 0.33 B	0

Note: In the table, uppercase letters indicate differences in longitudinal comparisons, which compare the number of different insect species at the same time point; identical letters indicate no significant difference (*p* > 0.05), while different letters signify a statistically significant difference (*p* < 0.05).

**Table 4 insects-16-00861-t004:** Comparative analysis of the trapping effects of different attractants and combinations on pests in QLAD species.

Period (Abbreviation)	Type/Combination of Chemicals	Pot Experiment		Field Experiment	
		Pest Species	Pest Quantity	Number of *T. angustipennis*	Male Adult Number of *Gynaephora* *menyuanensis*
Leaf unfolding stage (LS)	Ae	1 ± 0 a	5 ± 1.2 a	/	/
	Ae + Su	1 ± 0 a	6 ± 1.15 a	/	/
	Ae + Ch	1 ± 0 a	4.67 ± 0.33 a	/	/
	Ae + Su + Ch	1 ± 0 a	3.67 ± 0.33 a	/	/
Budding stage (BS)	Ae	2 ± 0.33 a	36 ± 4.33 a	/	/
	Ae + Su	2 ± 0.67 a	39 ± 4.26 a	/	/
	Ae + Ch	1 ± 0.33 a	46 ± 6.24 a	/	/
	Ae + Su + Ch	1 ± 0 a	51 ± 3.48 a	/	/
Initial flowering stage (IS)	Ae	1 ± 0 a	42 ± 5.29 b	45 ± 0.67 b	90 ± 1.53 b
	Ae + Su	2 ± 0.67 a	42 ± 6.08 b	48 ± 1.76 b	95 ± 3.84 b
	Ae + Ch	1 ± 0.33 a	65 ± 5.03 a	59 ± 1.33 a	101 ± 2.96 a
	Ae + Su + Ch	1 ± 0.33 a	68 ± 3.76 a	58 ± 2.91 a	101 ± 3.48 a
Peak bloom stage (PS)	Ae	1 ± 0 a	49 ± 9.06 b	47 ± 6.67 a	86 ± 14.93 a
	Ae + Su	2 ± 0.67 a	77 ± 12.6 ab	64 ± 6.12 a	82 ± 6.36 a
	Ae + Ch	2 ± 0.33 a	81 ± 7.22 a	52 ± 6.64 a	90 ± 7.84 a
	Ae + Su + Ch	2 ± 0.88 a	107 ± 4.18 a	57 ± 9.94 a	102 ± 15.13 a
Fruiting stage (FS)	Ae	1 ± 0 a	16 ± 0.88 b	35 ± 3.33 b	83 ± 9.02 a
	Ae + Su	1 ± 0 a	16 ± 0.33 b	46 ± 7.75 ab	89 ± 4.58 a
	Ae + Ch	1 ± 0 a	17 ± 1.15 b	65 ± 8.95 a	93 ± 11.32 a
	Ae + Su + Ch	1 ± 0 a	20 ± 0.58 a	58 ± 5.51 a	94 ± 13.3 a
Overall comparison of all periods (ALL)	Ae	2 ± 0.58 a	148 ± 17.01 c	127 ± 5.21 b	259 ± 24.58 a
	Ae + Su	3 ± 0.88 a	181 ± 15.04 bc	157 ± 5 a	266 ± 14.42 a
	Ae + Ch	2 ± 0.58 a	213 ± 12.44 ab	176 ± 11.62 a	285 ± 17.44 a
	Ae + Su + Ch	1 ± 0 a	249 ± 3.84 a	174 ± 8.19 a	296 ± 26.44 a

Note: The data in the table are presented as means ± standard deviations. The lowercase letters next to the numbers indicate differences in vertical comparisons (effectiveness of different types/combinations of agents during the same period). The same letter indicates no significant difference (*p* > 0.05), while different letters indicate a significant difference (*p* < 0.05).

## Data Availability

Due to the project not being completed, all data generated and analyzed during the current study are available from the corresponding author on reasonable request.

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
