# Peer review of "Ecological Pest Control in Alpine Ecosystems: Monitoring Asteraceae Phytophages and Developing Integrated Management Protocols in the Three River Source Region"

_insects, 2025, doi:10.3390/insects16080861_

Round 1
Reviewer 1 Report
Comments and Suggestions for Authors
Dear,
The manuscript entitled ‘Ecological Pest Control in Alpine Ecosystems: Monitoring Asteraceae Phytophages and Developing Integrated Management Protocols in the Three River Source Region’ has an interesting subject and worthy achievements. Management of Tephritis angustipennis using ecological-based methods is the main objective of this research. Please see the following comments, enhancing its quality:
Title: Asteraceae or Aster genus?
The cited references in the text do not follow the journal's instructions. Check them.
The abstract and introduction are very well written.
Lines 131-133: How was the species identified? With which key or by whom?
The result presentations, data analyses, discussion, and conclusion are excellent.
All the best
Author Response
Thank you very much for your review and feedback on my manuscript. For the detailed response to your comments, please refer to the attachment.

Reviewer 2 Report
Comments and Suggestions for Authors
The study puts forward a theoretical and practical framework for the management of arthropod pests associated with Asteraceae in the Three River Source Region located in the northeast of the Qinghai-Tibetan Plateau, China.
Overall, it investigates the ecology (species identification, population dynamics), physiology, and control of arthropod pests in the alpine environment. In relation to pest management specifically, it measures the effect of implementing three major IPM components: physical control (flower head bagging, lure-based trapping) and targeted chemical control, in suppressing arthropod pest populations. The implementation of the proposed IPM strategy will contribute to the ecological restoration of degraded alpine grasslands in this region.
General comments:
Overall, the topic is relevant, and the manuscript is well organized. However, in my view it is way too comprehensive and needs some revision for clarity, conciseness and wording. Scientific writing is about writing less, not more than necessary. Suggestion to consider removing all unnecessary adjectives and avoid wordiness. Perhaps some information could also be removed for conciseness.
I see this as at least two or three separate publications, one dealing with the ecological aspects of the study (population dynamics, etc.) and the other two with all other aspects concerning insect physiology and pest management, respectively.
To this reviewer, the way experimental set ups are described, especially in section 2.1, makes the writing hard to follow, please revise. I also would suggest physiological procedures, analysis and wording (sections 2.4 and 3.4) to be reviewed by an insect biochemistry specialist.
Specific comments:
Introduction
Line Comment
38 ‘synergistic control plan’. ‘Synergy’ involves interaction between components. I don’t see any specific analysis (Results, section 3.3) addressing the combined effect of two or more control methods versus each of them individually, so I’m not sure there is synergy in this case.
73 ‘focuses’ instead of ‘has largely focused’
82-85 ‘..these threats’- What threats? I assume it refers to the threat of pests reducing yields, not to the knowledge gaps?
89 The statement upon which ‘traditional’ chemical control is gradually being replaced by new control options but ‘still holds considerable importance’, I think needs stronger evidence. Please revise citations.
96-97 Delete ‘but’. My understanding is that research is ‘insufficient’ (?) and (chemical control) maybe ‘inadequate’ (‘ineffective’?) or rather ‘impractical’ for the TP grassland.
99 Any specific reference on ‘degradation of chemical control efficacy due to ultraviolet radiation’ would be helpful.
101 ‘.. also requires validation’. It is unclear to me why ‘also’. Does chemical control require validation?
104-108 Regardless of whether these are valid questions, are these the questions the study is aiming to address?
115-118 I do not think it is necessary to mention all the enzymes here but in the ‘Methods’ (section 2.4).
16, 32, and 120 ‘..an environmentally friendly IPM system’ sounds redundant. IPM is by definition an environmentally sound approach to pest management.
119-124 Combine first and second sentences. Line 124: ‘..in the context of global climate change’, though in my view the phrase could be removed. Again, avoid excessive verbiage.
Materials and Methods
137 Are these 30 plots outside of the 10 sample plots mentioned above? Or are these 30 ‘plants’ instead? Are there three ‘communities’ per sample plot thus making 30 ‘plots’ total? Please clarify.
138 50 ‘sweeps’ (..) 150 ‘sweeps’, instead of ’50 nets’.
164 Are QLAD, DRAS, MQAF, etc., plots? sites? plant species? Explain for clarity.
165-166 ’24 bagging treatments’; are the treatments different lure concentrations/timing of application? If not, please be clear on what treatments these are. I haven’t seen Table S2 but I think a Table showing this information is needed.
172-173 Please re-phrase. It is unclear to me how the 15 bottles total have been calculated.
210 Please be explicit on the indicators associated with PDR.
Results
Tables 1 and 2: Left column corresponds to ‘Sampling sites’ not ‘Species’. Also, consider using ‘growth period’ (LS, BS, IS, etc.) as column headers instead of ‘Sampling time’ and include the sampling times in the footnote.
271 ‘..with marked interspecific distribution differences..’. I would just say ‘marked differences in species [distribution] (‘abundance’?) within plots’. Similarly, in line 273: ‘T. angustipennis [showed] interannual consistency across plots [and years].
274 ‘..Ta densities than other [plots] (‘plots’ or ‘sites’?)
277-278 This is interesting; however, Fig. 3 does not say anything about differences in altitude across sites.
298 ‘inter-plot’ or ‘inter-site’ variations?
313-314 Note statement is hard to understand, please re-phrase. ‘[Species] were not surveyed at the T1 time point in either year’
331-332 This statement needs to be backed by experimental data
333 The subtitle in section 3.3 is misleading as it sounds as if there is some measurement of the combined effect between control methods (i.e physical versus targeted lure applications, etc.)
Fig. 4-F Why is CE plotted against ‘processing time’ (T1.., D1..) (as opposed to ‘spraying period’) and what does ‘D’ stand for in this graph?
Discussion
Suggestion to start the discussion with a concise, strong statement about the main contribution of the study.
461 Any reason why grazing pressure elevates Miridae abundance?
465-467 This statement needs to be backed by experimental data
505 ‘QLAD activates resistance enzymes..’, also in insects (lines 516-517): ‘defense’ enzymes instead of ‘resistance’ enzymes?
517-518 I wonder how priming the jasmonate pathway can enhance the timeliness of control. If it is by measuring the enzymatic activity to anticipate control actions, this may need further clarification.
Conclusions
576-580 I think this must be explicitly stated in the Results (section 3.3), including the experimental data. It must be clear to the reader where these results come from. Also, since this is about natural control, the word ‘utilization’ in lines 19 and 580 should be removed.
Author Response

(The authors gave the same response as above.)

Reviewer 3 Report
Comments and Suggestions for Authors
Dear authors,
Your manuscript is quite good in terms of the work done and the volume of information and data. And the subject is interesting and complexly approached.
However, some improvements are needed (listed below) to increase the value of your work.
All of these are also found in the manuscript directly in the right place.
Line 113-114: Related to purpose and objective 2) <evaluating the comprehensive efficacy of integrated control strategies combin-113 ing chemical regulation with physical measures; >
It is not clear whether objective 2 refers to all pests established within the diversity or only to the key ones and which ones are they? So please clarify for each objective which is the target organism(s). In the abstract you say control efficiency against Tephritis angustipennis while in the Introduction you generically mention only fruit flies and in the Objectives you no longer have target organisms.
Line 127:-128 : Related to: <The study was conducted from June 2023 to September 2024 in the TRSR of Qinghai 127 Province, China (altitude 2,303–4,115 m), based on previous research (Qin et al., 2023)>
Here you should be careful because saying it is...based on previous research may mean that you have taken over data and information, which is not right. Please clarify if some researchers started the study and you continue it and do not mention that you based on research to avoid overlapping ideas and results.
Line 128:-129 : Related to: < Ten sample plots were selected, including ...>
What was the size of each plot? Please mention it somewhere in the text.
Line 246: In Figure 2
Please replace the multiple graphics with clearer ones or arrange them 2 and 1 enlarged before making them an image so that they are readable. In its current form, even when enlarged, the characters are not clearly visible.
Line 260 : Related to: < ... causing severe damage to inflorescences of Aster spp...>
What are the damages? Who causes them and how is the plant affected? Please elaborate. It is important to provide information about this key pest. You did not even describe the species in the introduction, so you need to mention more somewhere.
Line 246: In Figure 6
Please enlarge the characters before making it an image. Not much is understood. When enlarging the image, no legible details are provided.
Lines 432-456: To Discussion
Summarize the Discussions to a maximum of 1.5 pages without subdivisions. You have already said a lot about the results. The Discussions should not be redundant and boring but should conclude your results fluently (as the basic ideas addressed) by comparing them with other approaches and results.
Kind regards,
R

Author Response

(The authors gave the same response as above.)

Round 2
Reviewer 2 Report
Comments and Suggestions for Authors
After carefully reading the cover letter and the revised version of the manuscript, I acknowledge that the manuscript has been sufficiently improved to warrant publication in 'Insects'.